# Effects of Noni (*Morinda citrifolia* L.) Fruit Extract Supplemented in Cashmere Goats with a High-Concentrate Diet on Growth Performance, Ruminal and Colonic Fermentation and SARA

**DOI:** 10.3390/ani13203275

**Published:** 2023-10-20

**Authors:** Qingyue Zhang, Shuhui Dong, Hao Yu, Yinhao Li, Xiaoyu Guo, Yanli Zhao, Yongmei Guo, Sumei Yan

**Affiliations:** Key Laboratory of Animal Nutrition and Feed Science at Universities of Inner Mongolia Autonomous Region, College of Animal Science, Inner Mongolia Agricultural University, Hohhot 010018, China; alicezqy@126.com (Q.Z.); d1978202906@126.com (S.D.); y18586266184@126.com (H.Y.); lucky.lyh@foxmail.com (Y.L.); gxy_2594@163.com (X.G.); ylzhao2010@163.com (Y.Z.); ymguo2015@163.com (Y.G.)

**Keywords:** noni, flavonoid, growth performance, nutrient digestibility, ruminal fermentation, colonic fermentation, cashmere goat

## Abstract

**Simple Summary:**

At present, cashmere goats are usually raised in a semi-intensive system to reduce the damage to grassland resources and ensure a more balanced intake of nutrients in China. Additionally, for maximizing production performance, intensive feeding management usually significantly increases the proportion of concentrate in the diet, which will seriously harm ruminant health in the long run. Noni fruit and its extracts showed good production-promoting and anti-inflammatory effects in poultry and freshwater fish production. However, it is unclear whether ethanol extract of noni fruit can alleviate the adverse effects of high-concentrate on goats. This study was conducted to investigate the effects of a high-concentrate diet supplemented with noni fruit extract (NFE) on growth performance, ruminal and colonic fermentation, nutrient digestion, and subacute rumen acidosis (SARA) of cashmere kids. The results showed that although a high-concentrate diet improved the growth performance of cashmere goats, it was at the expense of healthy fermentation modes of rumen and colon, such as significantly decreased pH and obviously increased lactic acid in rumen and colon. The supplementation of 0.10% NFE in a high-concentrate diet could not only effectively alleviate SARA and colon fermentation disorders induced by high-concentrate, but also improve the feed conversion efficiency in cashmere kids.

**Abstract:**

This experiment was conducted to investigate the effects of noni fruit extract (NFE) on growth performance, ruminal and colonic fermentation, nutrient digestion, and subacute rumen acidosis (SARA) of cashmere goats with the high-concentrate diet. Twenty-four cashmere kids (17.9 ± 1.45 kg of BW ± SD) were randomly assigned to three treatments: low-concentrate diet, high-concentrate (HC) diet, or HC diet supplemented with NFE at 1 g per kg DM (0.1%). The results showed that although the HC diet improved the average daily gain (ADG) and feed conversion rate (FCR), it was accompanied by SARA with a decreased pH and an increased lactic acid of both rumen and colon, and decreased digestibility of neutral detergent fiber (NDF)and acid detergent fiber (ADF). The supplementation of 0.10% NFE in the HC diet could not only effectively alleviate SARA symptoms and colon fermentation disorders, such as reversing the decrease of pH and alleviating the increase of lactic acid in rumen and colon, but also mitigate the decline of fiber digestibility caused by long-term feeding in the HC diet, and increase the digestibility of crude protein(CP) and dry matter (DM), which improved the ADG and FCR of cashmere kids. Thus, NFE provides new strategies for alleviating SARA and promoting cashmere goat growth.

## 1. Introduction

Inner Mongolia Cashmere goat with cashmere and meat dual-purpose breeds have a high reputation in China, mainly distributed in Ordos City and surrounding areas in Inner Mongolia, China. At present, cashmere goats are usually raised in a semi-intensive system to reduce the damage to grassland resources and ensure a more balanced intake of nutrients in China. In addition, in order to maximize the benefits, intensive feeding management usually significantly increases the proportion of concentrate in the diet to shorten fattening time and increase fattening weight [1,2]. Although high-concentrate diets can improve animal production efficiency in the short term, the long-term feeding of high-concentrate diets can cause systemic complications, such as subacute rumen acidosis (SARA) [3], imbalance of colon environmental homeostasis [4], liver damage [5], etc., which not only seriously endangers the health of ruminants, but also ultimately reduces economic benefits, and addresses a range of animal welfare and environmental issues.

With a long history of edible and medicinal use in its native Polynesia, the tropical plant noni (*Morinda citrifolia* L.) is also widely cultivated in southern China, including Guangdong, Hainan, and Taiwan. Secondary metabolites in noni fruit have a variety of pharmacological activities, such as antioxidant [6], anti-inflammatory [7], anticancer [8], and immune-boosting effects [9], which have attracted extensive attention from researchers. Noni fruit and its extracts showed good production-promoting and anti-inflammatory effects in poultry and freshwater fish production. The extract of noni fruit could improve the performance and blood parameters of freshwater fish with fluorine-induced poisoning [10]. Noni juice could improve growth performance, feed conversion efficiency, and blood variables in 120-day-old Sentul chickens [11]. Noni fermented juice could alleviate growth performance decline, oxidative stress, and immunosuppression of yellow catfish caused by a high-fat diet [6]. Dietary supplementation with different doses of noni fruit powder or its extract could improve heat stress, production performance, and feed conversion efficiency of broilers to varying degrees [12]. As important pharmacological active ingredients in noni fruit, polyphenols and flavonoids have various effects such as anti-oxidation [13,14], alleviating intestinal injury [15,16], and treating canker ulcer [17]. At present, the application of flavonoids or polyphenols from noni fruit in the production of animals, especially ruminants, is still lacking in relevant data reports. Other plant flavonoids and phenols showed good production-promoting properties in poultry [18], rabbits [19], and ruminants [20,21].

Given the performance-enhancing effects of noni and other plant flavonoids and polyphenols in livestock rearing, as well as the antioxidant and anti-inflammatory activities of noni active compounds, we hypothesized that noni ethanol extract could alleviate the adverse effects of high-concentrate feeding and promote growth performance in cashmere kids. Therefore, this trial was conducted to investigate the effects of the high-concentrate diet supplemented with ethanol extract from noni fruit on growth performance, ruminal fermentation, colonic fermentation, and nutrient digestion of cashmere goats, to develop novel feed additives for cashmere goats and provide a basis for the utilization of noni resources. It also provides new strategies for mitigating SARA.

## 2. Materials and Methods

The cashmere goat trial was carried out at the Inner Mongolia Agricultural University Science and Technology Park (40°40′ N; 111°23′ E), Hailiutu Village, Tumert left Banner, Hohhot, Inner Mongolia, China. According to the Laboratory Animal Sciences and Technical Committee of the Standardization Administration of China, the use of cashmere goats was approved by the Animal Ethics and Welfare Committee of Inner Mongolia Agricultural University, and conducted under the national standard Guidelines for Ethical Review of Animal Welfare [22].

### 2.1. Preparation of Ethanol Extract of Noni Fruit

In this trial, ripe noni fruit was from a noni fruit planting base in Wuzhishan, Hainan, China. An electric pulverizer (CH-200A, Chenhe Shengfeng Industry and Trade Co., Ltd., Yongkang, China) was used to crush the noni slices (dried at 65 °C) into powder and pass through a 1 mm screen. The ethanol extract of noni fruit (NFE) was obtained by reference to the ultra-sonic-assisted extraction method of Cui et al. (2017) [23] with a slight modification. In brief, noni fruit powder and ethanol/water (90:10, *v*/*v*) were mixed (1:35 ratio of raw powder to 90% ethanol solution, *w*/*v*) and extracted with ultrasonication at 70 °C for 15 min using ultrasonic equipment (DTC-27J, Dingtai Biochemical Technology Equipment Manufacturing Co., Ltd., Huanggang, China). After filtration (using filter paper with a maximum pore size of 15–20 μm), the filtrate was evaporated to 60 mL using a rotary evaporator (EYELA OSB-2200, Tokyo Riken Instrument Co., Ltd., Beijing, China) at 55 °C. Using a freeze dryer (Biosafer-10 C, Safer (China) Co., Ltd., Nanjing, China) to lyophilize the concentrated solution and obtained NFE (the yield was 27.1%), in which the structures of the active ingredients were identified by the ultraperformance liquid chromatography (UPLC; CBM30A, UFLC, Shimadzu, Kyoto, Japan) tandem mass spectrometry (ESI-MS, 6500 Q TRAP, AB Sciex Pte. Ltd., Shanghai, China) system [24]. The active ingredients of NFE are shown in Table 1.

### 2.2. Rearing Experiment

#### 2.2.1. Animals, Diets, and Feeding Management

In total, 24 healthy, 4-month-old, male Inner Mongolia Albas white cashmere goats (body weight = 17.9 ± 1.45 kg, mean ± SD) were selected and randomly assigned to one of three experimental treatments: high-concentrate (HC) diet, low-concentrate (LC) diet, or HC diet with supplementation with 1 g/kg DM NFE (0.1% NFE, HN). Each treatment had 8 goats. The supplementation level of NFE was determined by preliminary results (unpublished data, Zhang Q et al., Inner Mongolia Agricultural University, China) in our team. This trial lasted 14 weeks, including 2 weeks for adaptation and 12 weeks for treatment, which was divided into early (week 1–6) and late (week 7–12) periods to meet the different needs of different growth stages of cashmere goats for nutrients, according to the feeding standard of Nutrient Requirements of Meat-Type Sheep and Goat (NY/T 816-2021, Table 2) [25]. All feed ingredients were mixed in TMR except for NFE. Overall, 5 g of NFE was dissolved in 100mL 0.5% NaHCO_3_ solution (NaHCO_3_ for solubilization) to obtain NFE solution. A certain volume of NFE solution was mixed with 200 g TMR of HC diet according to supplementation of NFE at 1 g per kg DM, and top-dressed on the rest of the TMR in morning feeding. The corresponding volume of 0.5% NaHCO_3_ solution was added to the diet of HC goats and LC goats, respectively. The last 8 days of each period were for sampling and measurement. The treatment diets were offered to kids twice daily at 07:00 and 15:00. The goats were placed into individual pens (the area = 2.4 m^2^) with soil floor, and they had an individual feeder and water access. Deworming and stomach-strengthening were carried out before the adaptation period, and the pens were cleaned and disinfected regularly.

#### 2.2.2. Sampling and Measurements

The goats were weighed continuously before morning feeding on the first two days at the beginning and the last two days at the end of the early period and the late period. The average daily weight gain (ADG) and feed conversion ratio (FCR) of each period and the whole period were calculated. The feed was provided according to remaining refusals amount at 5–10%. By weighing and recording the amounts of feed supplied and the refusals, the intake was calculated. After dried at 65 °C for 48 h in an oven, samples of feed and refusals were ground using a pulverizer and passed through a 1 mm screen. Both in the early and late period, all cashmere kids were fitted with a fecal collection bag to collect and record the weight of daily output feces. In total, 20% of wet weight of feces were sampled and stored at −20 °C for subsequent analysis. During the measurement period, after thawed and mixed feces samples from 6 single days, the subsample was dried at 65 °C for 72 h, then ground to pass through a 1 mm screen.

During the last 2 days of the 8-day sampling period in the early period, ruminal contents (primarily the liquid phase) were collected using a gastric tube at 3 h after morning feeding (10:00). To prevent cross and saliva contaminations, the first 100 mL of each sample were discarded. Before next collection, the gastric tubes were washed and steeped using warm water [26]. All goats were slaughtered 3 h following the final morning feeding at the end of the trial. All rumen contents were poured out and mixed well. The pH of the above rumen contents collected at weeks 6 and 12, respectively, were determined immediately after collected by a portable pH meter (CT-6023; Kedida Electronics Co., Ltd. Shenzhen, China). Two aliquots samples were mixed with 3.5% formalin containing 0.6 g/L methyl green and 8.0 g/L sodium chloride (1:4, *v*/*v*) for microscopic counting of protozoa and stored at 4 °C; two aliquots of filtrate were mixed with 25% metaphosphoric acid solution (4:1, *v*/*v*) for analyzing volatile fatty acids (VFA); for analyzing ammonia-N (NH_3_-N), two aliquots of filtrate were mixed with 0.2 mol/L hydrochloric acid (1:9, *v*/*v*) to fix nitrogen; the residual ruminal fluid samples were collected for determination of microbial protein (MCP). Samples for VFA, NH_3_-N, and MCP were stored at −20 °C until analysis. The colon was exposed after slaughter and the colonic digesta samples were collected. The pH of digesta was determined immediately, then the samples were stored at −20 °C for subsequent analysis of lactate and VFA.

### 2.3. Determination of Chemical Ingredinents and Calculation of Apparent Nutrient Digestibility

According to the methods by the Association of Official Analytical Chemists [27], samples of feed and feces were analyzed for ether extract (EE) (method 973.18), CP (method 976.05), DM (method 930.15), phosphorus, and calcium (method 935.13). Using the Ankom 200I Fiber Analyser (Ankom Technology Co., New York, NY, USA) to analyze neutral detergent fiber (NDF) and acid detergent fiber (ADF), according to methods of Van Soest et al. [28], residual ash was removed using sodium sulfite and heat-stabilizing α-enzymes.
Apparent nutrient digestibility = [DMI (kg) × diet nutrient content (%) − fecal output (kg) × fecal nutrient content (%)]/DMI (kg) × diet nutrient content (%)

### 2.4. Determination of Ruminal and Clonic Fermentation Variables

The determination methods of VFA, NH_3_-N, MCP, and protozoa were the same as the most recent report in our team [29]. Briefly, a solution of 250 g/L metaphosphoric acid containing 2 g/L ethyl 2-butyrate was added to the supernatant obtained by centrifugation of rumen content samples (1:5, *v*/*v*). After mixing and centrifugation, the supernatant was used to analyze VFA by gas chromatography (GC-2014ATFSPL, Shimadzu, Kyoto, Japan; injector temperature 220 °C; detector temperature, 250 °C; column temperature, 180 °C; film thickness of the capillary column, 0.25 mm × 0.50 μm × 60 m). Total VFA (TVFA), molar percentage of each VFA in TVFA, and acetate to propionate ratio (A/P) were calculated. The number of ruminal protozoa were counted by microscope according to Dehority [30]. Using the phenol hypochlorite colorimetric method [31], the concentration of NH_3_-N was determined. According to the method of Makkar, the content of MCP was measured [32]. Using commercial antioxidant kits (Nanjing Jiancheng Bioengineering Institute of China, Nanjing, China), lactate content in the ruminal fluid was determined. In addition, 2 g of thawed colon digesta was mixed with 0.9% normal saline (1:1, *w*/*v*), then centrifuged at 2500× *g* for 15 min at 4 °C to obtain supernatant for determination of VFA and lactate contents. The detection method was the same as rumen fluid.

### 2.5. Statistical Analysis

For ruminal fermentation variables, the data were analyzed using PROC MIXED of SAS (version 8.1, SAS Institute Inc., Cary, NC, USA), which include the overall mean, fixed effect of diet, fixed effect of sampling period, fixed effects of the interaction of diet and sampling period, and residual error. Data of growth performance, nutrient digestibility, and colonic fermentation variables were also analyzed using PROC MIXED, removing the fixed effects of the sampling period and the interaction of diet and sampling period above. For the statistical analyses, *p* < 0.05 was declared as significant and 0.05 < *p* < 0.10 were considered as trends. The pdiff statement was used for multiple comparisons.

## 3. Results

### 3.1. Growth Performance

As presented in Table 3, compared with cashmere kids in LC, the final body weight was higher (*p* = 0.001) in HC and HN. During week 1–6, the ADG of kids in HC and HN were higher (*p* < 0.01) than that in LC, and ADG between HC and HN was similar. During weeks 7 to 12 and the whole trial, the ADG of kids in HC and HN were higher than that in LC, and in HN it was also higher than HC (*p* < 0.05). Compared with LC kids, the DMI in HC kids was increased (*p* = 0.016) during week 1–6, and the DMI in HN kids was increased (*p* = 0.004) during week 7–12. There was no significant difference among the three treatments in DMI during week 1–12. Compared with LC cashmere kids, the FCR in kids of HC and HN was increased during week 1–6, 7–12, and 1–12, and it was also higher in HN kids than in HC during week 7–12 and 1–12 (*p* < 0.05).

### 3.2. Nutrient Digestion

Table 4 shows the apparent nutrient digestibility of different treatments. The digestibility of DM in cashmere kids of HC and HN was higher than that in LC kids (*p* < 0.001), while the digestibility of CP in HN kids was higher than that in HC and LC kids (*p* = 0.013). Compared with LC kids, the digestibility of EE in early and late periods, and NDF in the early period were increased in HC and HN kids (*p* < 0.05). At week 12, the digestibility of NDF in HC kids was decreased (*p* = 0.024) compared with LC kids, but there was no significant difference between HN and the other two treatments. The digestibility of ADF in HC kids was lower than that in LC and HN kids at week 12 (*p* = 0.010).

### 3.3. Ruminal Fermentation

#### 3.3.1. pH, NH_3_-N, MCP, Protozoa, and Lactate of Rumen Fluid

As shown in Table 5, TRT × Week interaction was not significant for ruminal fermentation variables other than MCP concentration (*p* < 0.001). For MCP, the highest value was for the NFE-containing diet at week 6 and the lowest value was for the LC diet at week 12. Rumen fluid pH in cashmere kids of HC and HN was lower than that in LC kids, and pH in HC kids was lower than that in HN (*p* < 0.001). The differences among the three treatments in week 6 were consistent with the above. In week 12, the pH of HC kids, less than 5.5, was lower than that of kids in HN and LC. The contents of NH_3_-N and MCP in rumen fluid in HN goats were higher (*p* < 0.05) than those in HC and LC goats, and the difference between HC and LC goats was not significant. The differences in NH_3_-N concentration among three treatments in both weeks 6 and 12 were consistent with the above, as well as the differences in MCP concentration in week 12. In week 6, the MCP content in rumen fluid in HC goats was lower than that in HN and LC goats. The number of protozoa and lactate content in rumen fluid in HC cashmere kids was higher (*p* < 0.05) than those in HN and LC goats, and the difference between HN and LC kids was not significant. The differences for those two variables among three treatments in both weeks 6 and 12 were consistent with the above. In addition, the sample period had effects on the number of protozoa and contents of MCP and lactate (*p* < 0.005). Compared with week 6, the concentration of MCP during week 12 was reduced, but the number of protozoa and lactate content was increased.

#### 3.3.2. Rumen Fluid VFA

As shown in Table 6, the TRT × Week interaction has no significant effect on the variables, except for the molar percentage of iso-valerate in TVFA (*p* = 0.040), for which, the highest value and the lowest value was, respectively, for the NFE-containing diet and the HC diet in week 12. Compared with the LC kids, the molar percentage of valerate and acetate in kids of HC and HN was reduced, and that of acetate in HC kids was also lower than that in HN kids (*p* < 0.05). The differences among the three treatments for the molar percentage of acetate in week 12 and for the molar percentage of valerate in week 6 was consistent with the above. At week 6, the molar percentage of acetate in HC and HN kids was also lower than that in LC kids, but the difference between HC and HN kids was not significant. At week 12, the molar percentage of valerate in HC kids was higher than that in LC kids, but there was no significant difference between HN and the other two treatments. Compared with the LC kids, the molar percentage of propionate in kids of HC and HN was increased, and that in HC was higher than that in HN (*p* < 0.001). The differences among treatments in week 6 and 12 were consistent with the above, respectively. Compared with the LC kids, the value of A/P in kids of HC and HN was reduced, and the value in HC was lower than that in HN (*p* < 0.001). Individually, the A/P value in kids of HC and HN was lower than that in LC either in week 6 or 12. Additionally, the sampling period had effects (*p* < 0.05) on all variables except the molar percentage of iso-butyrate, valerate, and iso-valerate. Compared with week 6, the value of A/P and the molar percentage of acetate and valerate during week 12 were reduced, but that of propionate, butyrate, and TVFA were increased.

### 3.4. Colonic Fermentation

The colonic fermentation variables are presented in Table 7. The pH of colonic digesta in cashmere kids of HC was lower than that in kids of LC and HN, and there was no significant difference between the latter two treatments (*p* < 0.001). Compared with HC kids, TVFA concentration and the molar percentage of acetate in TVFA of kids in LC and HN was lower (*p* < 0.05), but that of iso-butyrate, valerate and iso-valerate was greater (*p* < 0.05). There was no significant difference between HN and LC kids in the above variables. Compared with kids in LC, the lactate content of colonic digesta in kids of HC and HN was greater, and it in HC kids was greater than HN (*p* < 0.001).

## 4. Discussion

### 4.1. Effects of NFE on Growth Performance and Nutrient Digestion of Cashmere Kids in the High-Concentrate Diet

Rumen, as a unique digestive organ of ruminants, plays an important role in protein and energy supply of the animal body. In this experiment, the HC diet increased the final body weight and ADG, decreased the FCR, and improved the growth performance of cashmere kids compared with LC kids. This was probably related to the change in rumen fermentation pattern. The concentration of propionate in HC kids was higher than that in LC kids, and the A/P value was lower than in LC kids. Although the concentration of propionic acid in HN kids was not significantly different from that in LC kids, the A/P value in HN kids was lower than that in LC kids. In ruminant rearing, propionate is more effective in promoting growth than acetate [33]. Therefore, the change in rumen fermentation pattern in HC and HN kids probably promoted the improvement of production performance.

Some studies have shown that the apparent digestibility of organic matter increases with the increase in the ratio of concentrate [34]. In this trial, the increased digestibility of DM and EE in kids of HC and HN during the late period, and the increased digestibility of NDF in the early period were also consistent with this statement. The increase in EE digestibility provided more energy for rumen microorganisms and cashmere goats, which was beneficial to improve the growth performance of HC kids. While the digestibility of EE increased in HN kids, the digestibility of CP also increased, which ensured the carbon and nitrogen balance of nutrient metabolism in the body. This was likely to be one of the reasons why HN kids showed better growth performance and feed conversion efficiency compared with HC and LC kids. At present, there are many studies on noni fruit and its extracts used to improve the growth performance of fish and poultry [6,9,10,11], but there are still few data reports on its application in ruminants. Limited data suggested that noni fruit meal could be used as a feed ingredient for Holstein dairy cows. The rumen degradable protein in noni meal was higher than that in wheat bran in vitro, and diet supplemented with noni fruit meal up to 1.5% could increase C18:1 fatty acid level in milk [35]. In addition, feed conversion efficiency of weaned beef calves increased linearly with the increasing noni pulp addition, but the supplementation of noni pulp had no significant effect on DMI of weaned cattle [36]. This is also similar to the results of this experiment. Similar results have also been obtained from studies of other plant extracts in ruminants. Diet of lactating dairy sheep supplemented with polyphenol-rich extract from grape seed could increase the ADG of their kids [21]. Supplementation of astragalus root extract in the diet of weaned calves could increase ADG and final weight, and decrease the DMI and DMI:ADG [37].

Apparent digestibility can not only indicate the utilization of nutrients by the animal body but also reflect the gastrointestinal function of the animal. The NDF digestibility of cashmere kids in HC and HN was greater than that of LC kids at the early period of this trial. While at the late period, the NDF and ADF digestibility of HC kids was lower than that of LC kids, and there was no significant difference between kids of HN and HC. The normal digestibility in the early period of various nutrients in HC kids was probably due to the adaptation mechanism of rumen barrier function in ruminants to HC diets in a short period [38]. The decrease in fiber digestibility in HC kids during the late period in this study was probably due to the decrease in pH value, which inhibited the growth and reproduction of fiber-degrading bacteria [39], and ultimately reduced the degradation rate of structural carbohydrates. In addition, Ivan et al. (2005) [40] showed that with the increase of NDF content in the diet, the flow of NDF in the digestive tract decreased and the NDF digestibility in rumen increased. This is also consistent with the results of NDF digestibility in the late period of this trial. It is not difficult to find that the difference in NDF content between the HC diet and the LC diet was widened in the late period. Interestingly, NDF and ADF digestibility in HC and HN decreased significantly in the late period than in the earlier period, but DM digestibility increased significantly in late period. This was probably because the proportion of NSC in the HC diet increased in the later period [31.25% vs. 46.35% (=48.33% increase)]. NSC are easier to digest than structural carbohydrate. Therefore, although the DM digestibility in the HC and HN kids during the late period was greater than that in the early period; the digestibility of NSC were mainly improved. Additionally, dietary NSC increase in a certain range could cause the decrease of fiber digestibility [41]. At the same time, HC diet caused a significant decrease in rumen pH, which affected the ability of the cellulolytic bacteria to break down fiber [42]. In addition, the digestibility of ADF and NDF in HN cashmere kids was greater than that in HC group in different levels, indicating that NFE supplementation mitigated the decrease of fiber digestibility in HC diet. Some studies have shown that mulberry leaf flavonoids could improve the relative abundance of cellulolytic bacteria in rumen fluid [43], which might be the reason why NFE mitigated the decrease in ADF digestibility of HC goats in this experiment. The abundance of the ruminal bacterial community was not measured in this trial, so the mechanism of NFE increasing fiber degradation needs further investigation.

### 4.2. Effects of NFE on Ruminal and Colonic Fermentation of Cashmere Kids in the High-Concentrate Diet

Rumen fermentation variables are important indicators to reflect ruminal homeostasis. pH is a prerequisite for the proper function of rumen and its microorganisms. A large amount of carbohydrates contained in the HC diet will be fermented rapidly after entering the rumen, resulting in the rapid production of VFA. Lactate is also a product of the rumen fermentation process. Although the content of lactate is low, the ionization constant of lactate (pKa = 3.9) is much lower than that of VFA (pKa = 4.8), so the contribution of lactate to pH in the rumen is much greater than that of VFA. All the above processes cause a rapid decline in pH and increase the risk of SARA in ruminants. Some studies have suggested that rumen pH below 5.5 can be diagnosed as SARA [44]. In this study, at weeks 6 and 12, rumen fluid pH at 3 h after morning feeding was significantly decreased in HC cashmere kids, and was also lower than 5.5, and lactate concentration in rumen fluid was significantly increased, both of which were characteristics of SARA. The supplementation of NFE in HN kids alleviated the decrease in pH and the increase of lactate in the HC diet. Similar results were obtained in the study on citrus flavonoids, HC diet supplementation with citrus flavonoid extract decreased the lactic acid concentration and increased the pH, which was related to increasing the relative abundance of lactate-consuming microorganism *Selenomomas ruminantium* and *Megaesphaera elsdenii*, and its pure component poncirine also decreased the relative abundance of *Streptococcus bovis* in vitro [45]. Another study showed that supplementation with plant extracts containing turmeric, thymol, and yeast cell wall components (mannan oligosaccharides and beta-glucans) could reduce the number of protozoan and fungi, and increased the relative abundance of lactic acid utilizer *Selenomomas ruminantium* in rumen [46]. A recent study has shown that rutin can inhibit the growth of lactate-producing bacteria in rumen fermentation in dairy cows in vitro, thereby reducing the concentration of lactate [47]. NFE contains 9.24% flavonoids, 9.18% polyphenols and 4.53% polysaccharides. These suggests that these active ingredients probably affect the relative abundance of lactate-producing bacteria and lactate-consuming bacteria in rumen, thereby reducing lactic acid concentration and reversing the pH decline induced by HC diet. In addition, as a precursor of lactate, the decrease of propionic acid concentration in the HN group may also be the main reason for NFE to alleviate the increase of lactic acid and the decrease of pH.

HC diets generally have a high starch content, and high concentrations of starch promote protozoan growth and reproduction [48]. Because of the engulfment of starch granules, the production of lactic acid can be slowed down to a certain extent. Interestingly, although the number of protozoa in HC cashmere kids was higher than that in HN kids, the pH was still lower than that in HN lambs. On the one hand, the reason was probably that the addition of NFE inhibited the lactic acid-producing bacteria in the rumen of cashmere goats as mentioned above, thereby reducing the lactic acid concentration, and alleviating the rise of pH. On the other hand, there might be a niche replacement in the rumen microbiota of HN cashmere goats, which may compensate for the gap in pH buffering in the rumen caused by the decrease in protozoan population by increasing the abundance of other microorganisms that have the effect of engulfment and utilizing starch. A recently study of our team showed that diet supplemented with polysaccharide rich extract of noni fruit could replace protozoa in a niche by increasing the ratio of Firmicutes to Bacteroides, in order to replace its role of fiber decomposition [29]. However, the rumen microbiota abundance was not determined in this study, so further studies are needed to explore the metabolism of NFE in the rumen and its interaction with rumen microorganisms. Protozoa cannot use NH_3_-N, but feed on bacteria [49], so the increase in the number of protozoa in HC kids in this trial was likely to be the main reason for the decrease in MCP concentration in rumen fluid. At present, there is still a lack of reports on the effect of NFE on ruminal fermentation. Saponins can achieve an antiprotozoal effect by destroying protozoan cell membranes and making intracellular substances extravasate [50]. Results of UPLC-MS/MS analysis in this trial showed that NFE contained a variety of triterpene saponins [hederagenin-GlcA-Ara, cadambagenic acid, oleanolic acid-3-O-glucosyl(1→2)glucoside, etc.], which may be one of the reasons why NFE has a good antiprotozoal effect.

Due to the smaller granularity of concentrate than roughage, it has a faster flow rate in the rumen [51], while the small intestine of ruminants has limited digestion of nutrients [52], which may increase the amount of nutrients entering the hindgut, resulting in enhanced fermentation and even acidosis of the hindgut. A study has shown that after 12 weeks of HC diet (ratio of concentrate to roughage = 70:30), the colonic pH of Boer goats decreased markedly and the concentrations of lactate, acetate, propionate, butyrate, and TVFA in colon increased significantly [53], which is consistent with the changes of colonic pH, VFA, and lactate in HC goats in this trial. Similar changes in cecal VFA concentration were observed in cows fed HC diets for a long time [54]. The addition of NFE also alleviated the fermentation disorders of colonic acidosis caused by HC diet with obviously increased VFA concentration and lactate concentration, and significantly decreased pH. Wang et al. (2023) [15] found that phenolics from noni fruit could alleviate intestinal damage in high fat diet-fed mice by modulating the gut microbiota. Another study has shown that noni juice-fortified yogurt could increase the contents of total flavonoids and total phenols in yogurt, and could also ameliorate colon length and histopathological changes in the ulcerative colitis mice via improving the expression of anti-inflammatory factors and reducing the expression of pro-inflammatory factors in spleen [14]. NFE contains 18.42% flavonoids and phenolic acids, indicating that the alleviating effect of NFE on colonic fermentation disorders caused by HC was also probably related to the regulatory effect of these active substances on cytokines. However, the cytokine content and their gene expression were not measured in this study, so further verification is needed in subsequent experiments.

In this trial, NFE was added in the form of a solution contained 0.5% NaHCO_3_. In this test, NaHCO_3_ was used to dissolve NFE for solubilization purpose. NFE has a certain viscosity, which may cause uneven mixing when mixed directly in feed. Therefore, we chosen dissolve it in liquid and feed it in the form of spraying on the diet. However, the dissolution of NFE in water was not ideal. The commonly used co-solvents are DMSO, glycerol and NaHCO_3_, etc., NaHCO_3_ was chosen as the cosolvent because its solution has a weak alkali, which was conducive to the dissolution of ethanol extract. In addition, NaHCO_3_ is a commonly used feed additive that can be used in HC diets to relieve SARA. Therefore, the introduction of trace amounts of NaHCO_3_ does not cause toxic side effects in animals. Moreover, NaHCO_3_ as a cosolvent also has the advantage of low cost. The preliminary test of our team showed that the concentration of NaHCO_3_ of 0.5% could dissolve NFE better. In order to eliminate the buffering effect of NaHCO_3_ on rumen pH, the same amount of 0.5% NaHCO_3_ solution was added to the HC and LC diets. In actual production, NFE can be dissolved with 0.5% NaHCO_3_ and sprayed on feed for use. Whether there is a better method for its application form needs to go into further detail.

## 5. Conclusions

Although the HC diet improved the growth performance of cashmere goats such as ADG and FCR, it was at the expense of healthy fermentation mode of rumen and colon such as a significant decrease in pH and an obvious increase in lactate concentration in rumen and colon. The supplementation of 0.10% NFE could not only effectively alleviate the rumen SARA symptoms and colon fermentation disorders caused by HC diet, but also improve the FCR of cashmere kids compared with the HC kids. Therefore, the NFE is recommended to be included in cashmere goat diets.

## Figures and Tables

**Table 1 animals-13-03275-t001:** The ingredients and compound contents of NFE (%, DM basis).

Ingredients	Content, %
Lipids	16.27
Amino acids and derivatives	12.87
Alkaloids	11.19
Flavonoids	9.24
Phenolic acids	9.18
Terpenoids	8.15
Organic acids	7.87
Nucleotides and derivatives	5.33
Saccharides	4.53
Quinones	4.47
Coumarins	2.82
Vitamins	2.00
Aldehydes	1.06
Others	5.03

**Table 2 animals-13-03275-t002:** Ingredients and chemical composition in diets to feed cashmere goat kids (air-dry basis).

Item	1–6 Week		7–12 Week	
	LC	HC ^5^	LC	HC ^5^
Ingredients (%)				
Alfalfa hay	20.00	8.75	32.00	14.00
Corn stalk	24.00	10.50	16.00	7.00
Oat grass hay	36.00	15.75	32.00	14.00
Corn grain	1.00	11.50	10.80	38.70
Soybean meal	13.50	5.10	4.25	6.15
Bran	0.90	20.00	0.70	8.00
Corn germ meal	1.10	25.00	0.50	8.00
Soybean oil	0.70	0.70	1.50	1.50
Premix ^1^	0.50	0.50	0.50	0.50
Limestone	1.00	1.55	0.20	1.20
CaHPO_4_	0.90	0.25	1.15	0.55
NaCl	0.40	0.40	0.40	0.40
Total	100.00	100.00	100.00	100.00
Nutrient composition				
Digestible energy, MJ/kg dry matter basis (DM) ^2^	11.16	11.79	11.83	12.80
DM, %, air dry basis	93.39	92.10	93.39	92.10
Nonstructural carbohydrate (NSC), %, DM	23.68	31.25	31.40	46.35
Crude protein (CP), %, DM	14.00	14.08	12.01	12.02
Ether extract, %, DM	2.46	3.53	3.57	4.32
aNDFom ^3^, %, DM	46.48	40.13	43.16	30.36
ADFom ^4^, %, DM	26.97	17.96	25.91	14.46
Calcium, %, DM	1.26	1.10	1.06	1.07
Phosphorus, %, DM	0.40	0.42	0.41	0.41

^1^ The premix provided the following per kg of diet: Fe 0.04 g, Cu 0.008 g, Zn 0.05 g, Mn 0.03 g, I 0.3 mg, Se 0.3 mg, Co 0.25 mg, VA 6000 IU, VD3 2500 IU, VE 12.5 IU, VK3 1.8 mg, VB1 0.035 mg, VB2 8.5 mg, VB6 0.9 mg, nicotinic acid 22 mg, D-pantothenic acid 17 mg, VB12 0.03 mg, biotin 0.14 mg, folic acid 1.5 mg. ^2^ Digestible energy and NSC were calculated based on the ingredients of the diet and their digestible energy content. ^3,4^ Neutral detergent fiber and acid detergent fiber both expressed exclusive of residual ash. ^5^ Ingredients and chemical composition of HN and HC diet are the same, only 1 g/kg DM of NFE was added to the diet in HN kids.

**Table 3 animals-13-03275-t003:** The effect of NFE on growth performance of cashmere kids in the high-concentrate diet.

Item	LC	HC	HN	SEM	*p*-Value
Initial body weight, kg	17.8	17.9	17.8	0.548	0.999
Final body weight, kg	24.7 ^b^	27.2 ^a^	28.2 ^a^	0.578	0.001
ADG, kg					
1–6 week	70.9 ^b^	108 ^a^	118 ^a^	4.62	<0.001
7–12 week	92.3 ^c^	114 ^b^	129 ^a^	4.83	<0.001
1–12 week	81.5 ^c^	111 ^b^	123 ^a^	3.35	<0.001
DMI, kg/d					
1–6 week	0.734 ^b^	0.912 ^a^	0.806 ^ab^	0.039	0.016
7–12 week	1.140 ^a^	1.023 ^ab^	0.894 ^b^	0.037	0.004
1–12 week	0.924	0.966	0.863	0.029	0.056
FCR					
1–6 week	10.4 ^a^	8.59 ^b^	6.99 ^b^	0.609	0.003
7–12 week	12.2 ^a^	9.07 ^b^	7.26 ^c^	0.581	<0.001
1–12 week	11.3 ^a^	8.80 ^b^	7.10 ^c^	0.509	<0.001

^a,b,c^ Without a common superscript letter indicates significant (*p* < 0.05) changes between treatments.

**Table 4 animals-13-03275-t004:** The effect of NFE on apparent nutrient digestibility of cashmere kids in the high-concentrate diet (%).

Item	LC	HC	HN	SEM	*p*-Value
DM					
Week 6	60.1	59.4	60.3	1.163	0.854
Week 12	57.6 ^b^	66.7 ^a^	69.7 ^a^	1.606	<0.001
CP					
Week 6	71.9	72.4	69.5	2.713	0.727
Week 12	67.1 ^b^	68.0 ^b^	72.0 ^a^	1.133	0.013
EE					
Week 6	68.6 ^b^	78.9 ^a^	77.7 ^a^	2.708	0.027
Week 12	75.9 ^b^	83.2 ^a^	84.4 ^a^	1.075	<0.001
NDF					
Week 6	35.6 ^b^	52.2 ^a^	57.5 ^a^	2.828	<0.001
Week 12	45.3 ^a^	35.2 ^b^	40.8 ^ab^	2.391	0.024
ADF					
Week 6	48.7	42.2	47.4	3.118	0.324
Week 12	41.5 ^a^	28.2 ^b^	35.9 ^a^	2.785	0.010

^a,b^ Without a common superscript letter indicates significant (*p* < 0.05) changes between treatments.

**Table 5 animals-13-03275-t005:** The effect of NFE on ruminal pH, NH_3_-N, MCP, protozoa, and lactate of cashmere kids in the high-concentrate diet.

Item	Week 6			Week 12			Week		TRT			SEM	*p*-Value		
	LC	HC	HN	LC	HC	HN	6	12	LC	HC	HN		Week	TRT	TRT × Week
pH	6.17 ^a^	5.44 ^c^	5.76 ^b^	5.94 ^a^	5.35 ^b^	5.73 ^a^	5.79	5.67	6.06 ^A^	5.39 ^C^	5.74 ^B^	0.065	0.101	<0.001	0.509
NH_3_-N, mg/100 mL	11.6 ^b^	13.5 ^b^	23.6 ^a^	13.8 ^b^	9.14 ^b^	21.9 ^a^	16.2	15.0	12.7 ^B^	11.3 ^B^	22.7 ^A^	1.35	0.396	<0.001	0.200
MCP, mg/100 mL	30.3 ^a^	24.1 ^b^	31.1 ^a^	9.25 ^b^	11.2 ^b^	15.8 ^a^	28.5 ^A^	12.1 ^B^	19.7 ^B^	17.6 ^B^	23.4 ^A^	0.904	<0.001	<0.001	<0.001
Protozoa, 10^4^/mL	105 ^b^	209 ^a^	117 ^b^	151 ^b^	280 ^a^	128 ^b^	144 ^B^	186 ^A^	128 ^B^	245 ^A^	122 ^B^	12.6	0.006	<0.001	0.248
Lactate, mmol/L	0.556 ^b^	0.817 ^a^	0.614 ^b^	0.694 ^b^	1.090 ^a^	0.754 ^b^	0.662 ^B^	0.847 ^A^	0.625 ^B^	0.954 ^A^	0.684 ^B^	0.0519	0.004	<0.001	0.539

^a,b,c^ Without a common superscript letter indicates significant (*p* < 0.05) changes among treatments in the same week. ^A,B,C^ Without a common superscript letter indicates significant (*p* < 0.05) changes among treatments in the same week.

**Table 6 animals-13-03275-t006:** The effect of NFE on ruminal VFA of cashmere kids in the high-concentrate diet.

Item	Week 6			Week 12			Week		TRT			SEM	*p*-Value		
	LC	HC	HN	LC	HC	HN	6	12	LC	HC	HN		Week	TRT	TRT × Week
TVFA, mmol/L	46.7	49.6	50.6	61.3	63.7	57.6	49.0 ^B^	60.9 ^A^	54	56.6	54.1	2.28	<0.001	0.656	0.350
(mol/100mol)															
Acetate	69.7 ^a^	58.0 ^b^	61.1 ^b^	66.2 ^a^	52.7 ^c^	57.3 ^b^	62.9 ^A^	58.8 ^B^	68.0 ^A^	55.4 ^C^	59.2 ^B^	1.127	0.002	<0.001	0.801
Propionate	18.4 ^c^	33.0 ^a^	26.4 ^b^	18.4 ^c^	33.0 ^a^	26.4 ^b^	22.8 ^B^	25.9 ^A^	17.7 ^C^	30.7 ^A^	24.6 ^B^	1.499	0.033	<0.001	0.653
Butyrate	10.8	10.7	13.1	12.8	11.3	12.7	11.5	12.3	11.8	11	12.9	0.548	0.262	0.062	0.298
Iso-butyrat	0.706	0.568	0.68	0.666	0.458	0.75	0.652	0.625	0.686	0.513	0.715	0.071	0.579	0.117	0.325
Valerate	1.02 ^b^	1.42 ^a^	1.52 ^a^	1.26 ^b^	1.90 ^a^	1.67 ^ab^	1.32 ^A^	1.61 ^B^	1.14 ^B^	1.66 ^A^	1.60 ^A^	0.109	0.009	0.006	0.391
Iso-valerate	0.742	0.888	0.743	0.648	0.64	1.125	0.791	0.804	0.695	0.764	0.934	0.123	0.893	0.383	0.040
A/P	4.26 ^a^	2.16 ^b^	2.74 ^b^	3.63 ^a^	1.71 ^b^	2.35 ^b^	3.06 ^A^	2.56 ^B^	3.94 ^A^	1.94 ^C^	2.55 ^B^	0.180	0.013	<0.001	0.854

^a,b,c^ Without a common superscript letter indicates significant (*p* < 0.05) changes among treatments in the same week. ^A,B,C^ Without a common superscript letter indicates significant (*p* < 0.05) changes among treatments in the same week.

**Table 7 animals-13-03275-t007:** The effect of NFE on colonic fermentation variables of cashmere kids in the high-concentrate diet.

Item	LC	HC	HN	SEM	*p*-Value
pH	7.17 ^a^	6.62 ^b^	7.01 ^a^	0.066	<0.001
Total VFA, mmol/L	42.7 ^b^	76.8 ^a^	48.9 ^b^	3.738	<0.001
(mol/100 mol)					
Acetate	74.7 ^b^	77.2 ^a^	74.4 ^b^	0.655	0.013
Propionate	17.2	15.6	16.4	0.558	0.168
Butyrate	4.92	5.56	5.82	0.385	0.260
Iso-butyrate	1.08 ^a^	0.49 ^b^	1.29 ^a^	0.124	0.001
Valerate	1.22 ^a^	0.81 ^b^	1.16 ^a^	0.071	0.001
Iso-valerate	0.877 ^a^	0.345 ^b^	0.958 ^a^	0.111	0.004
A/P	4.36	5.01	4.60	0.224	0.147
Lactate, mmol/L	0.794 ^c^	1.39 ^a^	1.20 ^b^	0.059	<0.001

^a,b,c^ Without a common superscript letter indicates significant (*p* < 0.05) changes between treatments.

## Data Availability

The data presented in this study are available on request from the corresponding author.

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
