# Peer review of "Effects of Noni (Morinda citrifolia L.) Fruit Extract Supplemented in Cashmere Goats with a High-Concentrate Diet on Growth Performance, Ruminal and Colonic Fermentation and SARA"

_animals, 2023, doi:10.3390/ani13203275_

Round 1
Reviewer 1 Report
The title should be modified to reflect the main substance used (Noni fruit extract (NFE)).
The abstract should include values that were key findings in this experiment.
The authors touched on growth performance too briefly in the discussion; please include evidence from comparable research, notably on DMI, ADG, and FCR. It was critical if you wanted to encourage the use of NFE in goat diet.
The authors should elaborate on DMD of weeks 7-12 in HC and HN, which were clearly higher in weeks 1-6, while NFD and ADFD were in deposit way.
Rumen pH was higher in the HN group than in the HC group; the authors should explain why NFE can enhance rumen pH further. What were the active ingredients in NFE or NaHCO3 responsible for?
Protozoa levels were higher in the HC group than in the HN group, whereas rumen pH was lower in the HC group than in the HN group. In theory, high protozoa populations may benefit in the engulfment of starch granules, slowing lactic acid production; however, the authors did not explain why.
Any studies demonstrating that secondary compounds in NFE are responsible for inhibiting microorganisms involved in lactic acid production in the rumen should be cited by the authors.
In the conclusion section, the authors suggested employing NFE at 0.1%. According to the Materials and Methods, why did the authors dissolve NFE with 0.5%NaHCO3 while using NFE. Thus, the authors had to go into further detail about the use of NFE in animal diets.
Table 3 should include the initial weight of the goats.
Tables 5 and 6 should follow the same format as Tables 2–4.
Please modify the unit of VFA in Tables 6 and 7 from mmol/L to mol/100 mol to make it easier for readers to follow.
Author Response
Response to Reviewer 1 Comments
Dear Reviewer,
Thank you very much for your precious comments and advice. Those comments are all valuable and very helpful for revising and improving our paper, as well as the important guiding significance to our researches. We have studied comments carefully and revised our manuscript based on them. The detailed responds to your review are attached below.
We would love to thank you for allowing us to resubmit a revised copy of the manuscript and we highly appreciate your time and consideration.
Kind regards,
Authors
- The title should be modified to reflect the main substance used (Noni fruit extract (NFE))
au: Thanks for your suggestion. We have changed “ethanol extract from noni (Morinda citrifolia L.) fruit” to noni (Morinda citrifolia L.) fruit extract. Please see the title.
- The abstract should include values that were key findings in this experiment.
au: Thanks for your suggestion. We agree with you and have focused on describing the mitigation effect of NFE on HC diet induced SARA. We also have added a description of the mitigation effect of NFF on fiber digestibility decline caused by long-term feeding of HC diets and the improvement of CP and DM digestibility in the abstract section, as detailed in L34-41.
- The authors touched on growth performance too briefly in the discussion; please include evidence from comparable research, notably on DMI, ADG, and FCR. It was critical if you wanted to encourage the use of NFE in goat diet.
au: Thanks for your suggestion. We agree with you and we have provided the limited reports on the use of noni resources in ruminants, and added growth promotion data on other plant extracts in ruminants to enrich the discussion. Please see chapter L347~360.
- The authors should elaborate on DMD of weeks 7-12 in HC and HN, which were clearly higher in weeks 1-6, while NFD and ADFD were in deposit way.
au: Thanks for your suggestion. We agree with you. We have added the content of non-structured carbohydrate (NSC) to the dietary composition table (please see L139 Table2), and explained this phenomenon from the perspective of the significantly increased NSC content in the HC diet of the late period and the easier digestion of NSC than structural carbohydrate. Please see L375~387.
- Rumen pH was higher in the HN group than in the HC group; the authors should explain why NFE can enhance rumen pH further. What were the active ingredients in NFE or NaHCO3 responsible for?
au: Thanks for your suggestion. We agree with you. According to your suggestion, we have discussed in detail why NFE could enhance rumen pH further, as detailed in L408~416. Besides, we illustrate the active ingredients in NFE that may play a role, as detailed in L418~424.
- Protozoa levels were higher in the HC group than in the HN group, whereas rumen pH was lower in the HC group than in the HN group. In theory, high protozoa populations may benefit in the engulfment of starch granules, slowing lactic acid production; however, the authors did not explain why.
au: Thanks for your suggestion. We agree with you and we have added a discussion on this interesting phenomenon, as detailed in L427~442.
- Any studies demonstrating that secondary compounds in NFE are responsible for inhibiting microorganisms involved in lactic acid production in the rumen should be cited by the authors.
au: Thanks for your suggestion. We agree with you. According to your suggestion, we have revised the discussion of this part, as detailed in L408~416.
- In the conclusion section, the authors suggested employing NFE at 0.1%. According to the Materials and Methods, why did the authors dissolve NFE with 0.5% NaHCO3 while using NFE. Thus, the authors had to go into further detail about the use of NFE in animal diets.
au: Thanks for your suggestion. We agree with you. In fact, the Materials and Methods section of the previously submitted manuscript has indicated the use of NaHCO3 for solubilization (please see L130~131). According to your suggestion, we have discussed in detail the reasons for using NaHCO3 solution as a solvent in the discussion, and pointed out that whether there is a better way to use in actual production needs to go into further detail. Please see L473~487.
- Table 3 should include the initial weight of the goats.
au: Thanks for your advice. We agree with you and the initial weight has been added, as detailed in L224 Table 3.
- Tables 5 and 6 should follow the same format as Tables 2–4.
au: Thanks for your kind suggestion. However, reviewer 2 recommended that the mean values for week should be added in the table according to the definition of the two-way analysis. After modification according to above suggestion, due to the limited space, the above content cannot be put down according to the tabulation format as Table 2-4. Therefore, we used different tabulation formats for one-way analysis and two-way analysis. We hope you can understand.
- Please modify the unit of VFA in Tables 6 and 7 from mmol/L to mol/100 mol to make it easier for readers to follow.
au: Thanks for your suggestion. We agree with you. The unit of VFA has been revised, and the Results section has been modified accordingly. Please see L279~299, Table 6 (L300), L307~311, and Table 7 (L320).
Again, many thanks for your valuable reviews.

Reviewer 2 Report
Effects of high-concentrate diet supplemented with ethanol extract from Noni (Morinda citrifolia L.) fruit on growth performance, ruminal and colonic fermentation and nutrient digestion in cashmere goats
Dear Authors,
the manuscript is well prepared, but from point of statistical view the methodical part (2.5) and tables must be better explained (one-way or/and two-way ANOVA). Experiment shows that use of the ethanol extract brings very positive effects: alleviation of SARA symptoms and problems with fermentation process in colon. Additionally improvement of FCR cashmere kids was observed, what can also affect for its healthiness and mortality. Below I add some suggestions helpful during this process:
Line 116
In the experiment 8 blocks were constituted, but in total participated there 24 animals. Design of experiment is proper, but power of a tests could be very low. Maybe in future will be possible to increase power of a test from 8 replications in each treatment. To increase power of a test, many small blocks that contain one observation (replication) per treatment group 8 x 3 animals in small block = 24 animals in treatment and 3 treatments, that gives 24 x 3 = 72 animals.
Line 135
In text is: ‘…The treatment diets were offered to kids twice daily at 0700 and 1500 h…’. Must be: at 07:00 and 15:00.
Line 206
Information about the analysis of variance (ANOVA) is needed: one-way and two way, in this case despite the model of experiment for two-way (Table 5 and 6) one way (Table 3, 4 and 7) also can be added and information about post-hoc test (Duncan’s, Tukey’s, or other).
Line 230
Table 3
SEM values must have 3 decimals
DMI – majority of values have 3 decimals in this case DMI from 7th to 12th week must have equal number decimal (0 could be added as a third decimal).
Line 251
Table 3
SEM values must have 3 decimals
Last row (ADF, week 12) – letters in superscript needed.
Line 280
Table 3
SEM values must have 3 decimals.
This table is presents using the two-way ANOVA, but in first two blocks-rows you mean values, one-way ANOVA table are presented. From the definition of the two-way analysis, there should be present in the table mean values for TRT and week (in this case, value for week is calculated from all observations-replications obtained in 6th and 12th week (8+8) replications = 16 replications in total. One additional block-row after mean values for week factor after values for week 12 can be added, what will separate this part of table as a one factorial ANOVA and after that will be values for two-factorial ANOVA (Week and TRT as a factors, because in analysis you determined interaction between week and TRT, without separating for 6th and 12th week).
There is no need to change letters in case the two factorial ANOVA. Second factor can also have the same letters in the superscript (a, b, c; difference between mean values in treatments exist).
TRT, p-value, Protozoa and Lactate column, first zero before point disappeared, in those two cases.
Line 309
Table 6
SEM values must have 3 decimals.
This table is presents using the two-way ANOVA, but in first two blocks-rows you mean values, one-way ANOVA table are presented. From the definition of the two-way analysis, there should be present in the table mean values for TRT and week (in this case, value for week is calculated from all observations-replications obtained in 6th and 12th week (8+8) replications = 16 replications in total. One additional block-row after mean values for week factor after values for week 12 can be added, what will separate this part of table as a one factorial ANOVA and after that will be values for two-factorial ANOVA (Week and TRT as a factors, because in analysis you determined interaction between week and TRT, without separating for 6th and 12th week).
There is no need to change letters in case the two factorial ANOVA. Second factor can also have the same letters in the superscript (a, b, c; difference between mean values in treatments exist).
Valerate column - values must have the same number of decimals.
Line 336
Table 7
SEM values must have 3 decimals.
Line 494
Maybe better is to add link to website with source: https://animalsciencejournal.usamv.ro/index.php/scientific-papers/past-issues?id=631
Line 502
In this case link is available:
https://doi.org/10.9721/KJFST.2021.53.4.446
Line 574
Please check if DOI appears in case of this publication, if not reference could be added without link, like in case nutritional requirements.
Line 586
Link with DOI needed.
Author Response
Response to Reviewer 2 Comments
Dear Reviewer,
Thank you very much for your precious comments and advice. Those comments are all valuable and very helpful for revising and improving our paper, as well as the important guiding significance to our researches. We have studied comments carefully and revised our manuscript based on them. The detailed responds to your review are attached below.
We would love to thank you for allowing us to resubmit a revised copy of the manuscript and we highly appreciate your time and consideration.
Kind regards,
Authors
- the manuscript is well prepared, but from point of statistical view the methodical part (2.5) and tables must be better explained (one-way or/and two-way ANOVA). Experiment shows that use of the ethanol extract brings very positive effects: alleviation of SARA symptoms and problems with fermentation process in colon. Additionally improvement of FCR cashmere kids was observed, what can also affect for its healthiness and mortality. Below I add some suggestions helpful during this process.
au: Thank you for your recognition and positive comments on this study. We will do our best to address your concerns in order to make the manuscript more complete and more informative.
- Line 116 In the experiment 8 blocks were constituted, but in total participated there 24 animals. Design of experiment is proper, but power of a tests could be very low. Maybe in future will be possible to increase power of a test from 8 replications in each treatment. To increase power of a test, many small blocks that contain one observation (replication) per treatment group 8 x 3 animals in small block = 24 animals in treatment and 3 treatments, that gives 24 x 3 = 72 animals.
au: Thank you for your valuable advice. Your suggestions will be taken into account in subsequent animal feeding trials.
- Line 135 In text is: ‘…The treatment diets were offered to kids twice daily at 0700 and 1500 h…’. Must be: at 07:00 and 15:00.
au: Thank you for your suggestion. We have changed. Please see L133.
- Line 206 Information about the analysis of variance (ANOVA) is needed: one-way and two way, in this case despite the model of experiment for two-way (Table 5 and 6) one way (Table 3, 4 and 7) also can be added and information about post-hoc test (Duncan’s, Tukey’s, or other).
au: Thanks for your kind suggestion. We agree with your suggestion. In fact, the last line of the Statistical Analysis in the original manuscript has given the test information for the multiple comparisons used in this experiment. For multiple comparisons, we used the pdiff statement (please see L209~210), which has similar functionality to Duncan and Tukey.
- Line 230 Table 3 SEM values must have 3 decimals. DMI – majority of values have 3 decimals in this case DMI from 7th to 12th week must have equal number decimal (0 could be added as a third decimal).
au: Thanks for your suggestion. The decimal number of results in this paper is reserved according to three significant digits. We agree with your suggestion and have revised it, as shown in L224 Table 3.
- Line 251 Table 4 SEM values must have 3 decimals. Last row (ADF, week 12) – letters in superscript needed.
au: Thanks for your suggestion. We are very sorry that we did not set the letters in superscript due to our oversight. It has been revised, and the decimal number has also been revised, as shown in L243 Table 4.
- Line 280 Table 5 SEM values must have 3 decimals. This table is presents using the two-way ANOVA, but in first two blocks-rows you mean values, one-way ANOVA table are presented. From the definition of the two-way analysis, there should be present in the table mean values for TRT and week (in this case, value for week is calculated from all observations-replications obtained in 6th and 12th week (8+8) replications = 16 replications in total. One additional block-row after mean values for week factor after values for week 12 can be added, what will separate this part of table as a one factorial ANOVA and after that will be values for two-factorial ANOVA (Week and TRT as a factors, because in analysis you determined interaction between week and TRT, without separating for 6th and 12th week).
au: Thanks for your suggestion. We agree with your suggestion. SEM values have been changed, and the mean values of factor week have been added, as shown in L274 Table 5.
- Line 280 Table 5 There is no need to change letters in case the two factorial ANOVA. Second factor can also have the same letters in the superscript (a, b, c; difference between mean values in treatments exist).
au: Thanks for your kind advice. We agree with your suggestion. The letters have been changed to the same, as shown in L274 Table 5.
- Line 280 Table 5 TRT, p-value, Protozoa and Lactate column, first zero before point disappeared, in those two cases.
au: Thanks for your careful advice. We are very sorry for missing the zero before the decimal point due to our negligence. The zero has been added, as shown in L274 Table 5.
- Line 309 Table 6 SEM values must have 3 decimals. This table is presents using the two-way ANOVA, but in first two blocks-rows you mean values, one-way ANOVA table are presented. From the definition of the two-way analysis, there should be present in the table mean values for TRT and week (in this case, value for week is calculated from all observations-replications obtained in 6th and 12th week (8+8) replications = 16 replications in total. One additional block-row after mean values for week factor after values for week 12 can be added, what will separate this part of table as a one factorial ANOVA and after that will be values for two-factorial ANOVA (Week and TRT as a factors, because in analysis you determined interaction between week and TRT, without separating for 6th and 12th week).
au: Thanks for your suggestion. We agree with you. SEM values have been changed, and the mean values of factor week have been added, as shown in L300 Table 6. In addition, the unit was changed to mol/100mol as recommended by reviewer 1, please note. Changes have also been made in the Results, as detailed in L279~299.
- Line 309 Table 6 There is no need to change letters in case the two factorial ANOVA. Second factor can also have the same letters in the superscript (a, b, c; difference between mean values in treatments exist).
au: Thanks for your kind advice. We agree with your suggestion. The letters have been changed to the same, as shown in L300 Table 6.
- Line 309 Table 6 Valerate column - values must have the same number of decimals.
au: Thanks for your suggestion. We agree with you and the number of decimals has been changed, as shown in L300 Table 6. In addition, the unit was changed to mol/100mol as recommended by reviewer 2, please note.
- Line 336 Table 7 SEM values must have 3 decimals.
au: Thanks for your suggestion. We agree with you. SEM values have been changed, as shown in L320 Table 7.
- Line 494 Maybe better is to add link to website with source: https://animalsciencejournal.usamv.ro/index.php/scientific-papers/past-issues?id=631
au: Thanks for your kind suggestion. We have changed. Please see L548
- Line 502 In this case link is available:
https://doi.org/10.9721/KJFST.2021.53.4.446
au: Thanks for your kind suggestion. We are very sorry that we did not find the doi number of this article before. It has now been modified (please see L556). Thank you again for your kind reminder and providing a more accurate website.
- Line 574 Please check if DOI appears in case of this publication, if not reference could be added without link, like in case nutritional requirements.
au: Thanks for your kind advice. We are very sorry that the doi number of this article was not found before due to our negligence. It has been supplemented. Please see L643.
- Line 586 Link with DOI needed.
au: Thanks for your careful advice. We are very sorry for missing the Link with doi of this article due to our negligence. The link was supplemented. Please see L661.
Again, many thanks for your valuable reviews.

Round 2
Reviewer 1 Report
The authors had made numerous changes to the manuscript in order to make various parts more apparent. However, several issues must be solved.
The first two remarks were not appropriately addressed. The title should be changed to reflect the major component utilized (Noni fruit extract (NFE)). For instance, it ought to say "Effects of Noni (Morinda citrifolia L.) Fruit Extract supplemented in Cashmere Goats fed High-Concentrate Diet on Growth Performance, Ruminal and Colonic Fermentation and SARA" . In addition to the abstract, the authors did not include any scientific values such as ADG, FCR, or rumen pH but instead focused on SARA. If the authors intend to develop new techniques for minimizing SARA, they should present solid proof that SARA was the major issue that needed to be addressed.
NFE appears to contain many secondary compounds with multifunctional properties that aid rumen and hind gut fermentation and, as a result, increase growth performance. If that's the case, could you perhaps include a recommendation in the conclusion to include NFE in Cashmere goat diets?
Lines 82 and 88 should be changed from 'trail' to 'trial'.
Please correct the sentence in line 121.
Line 455: the concentrate-to-roughage ratio was 65:35 rather than 70:30.
Line 494: Do you mean HN or HC, or should you change the statement to avoid confusion?
Tables 5 and 6 were highly advised to use the same format as Tables 3, 4, and 7.
Please correct Table 7. Something went wrong, so please correct it.
Author Response
Dear Reviewer,
Thank you very much for your thoughtful suggestions that have helped improve this paper substantially. We apologize for the inappropriateness of the last revision. We have studied the comments carefully and revised our manuscript based on it. The detailed responds to your review are attached below.
We would love to thank you for allowing us to resubmit a revised copy of the manuscript and we highly appreciate your time and consideration.
Sincerely,
Authors
- The authors had made numerous changes to the manuscript in order to make various parts more apparent. However, several issues must be solved.
au: Thank you for your recognition and positive comments on last revision. We will do our best to address your other concerns in order to make the manuscript completer and more informative.
- The first two remarks were not appropriately addressed. The title should be changed to reflect the major component utilized (Noni fruit extract (NFE)). For instance, it ought to say "Effects of Noni (Morinda citrifolia L.) Fruit Extract supplemented in Cashmere Goats fed High-Concentrate Diet on Growth Performance, Ruminal and Colonic Fermentation and SARA".
au: Thanks for your advice. We agree with you and the title have been revised according to your instance. Please see the title for details.
- In addition to the abstract, the authors did not include any scientific values such as ADG, FCR, or rumen pH but instead focused on SARA. If the authors intend to develop new techniques for minimizing SARA, they should present solid proof that SARA was the major issue that needed to be addressed.
au: Thanks for your suggestion. We agree with you. In the abstract, the scientific values of the effects of the HC diet on, for example, ADG, FCR, pH and fiber digestibility have been added, as detailed in L33~34, so that the proof for the major issue SARA was more solid in this trial.
- NFE appears to contain many secondary compounds with multifunctional properties that aid rumen and hind gut fermentation and, as a result, increase growth performance. If that's the case, could you perhaps include a recommendation in the conclusion to include NFE in Cashmere goat diets?
au: Thank you for kind advice. The recommendation for NFE to be included in the cashmere goat diet has been added at the end of the conclusion. Please see L485~486 for details.
- Lines 82 and 88 should be changed from 'trail' to 'trial'.
au: Thank you for careful advice. We are very sorry that we have written the wrong word by mistake, and this word have been checked and corrected in the whole manuscript. Please see L82 and L88.
- Please correct the sentence in line 121.
au: Thanks for your careful advice. This sentence has been revised, as detailed in L121.
- Line 455: the concentrate-to-roughage ratio was 65:35 rather than 70:30.
au: Thanks for your advice. Are you referring to the description of citation 55? We have verified the original article and they did use the ratio of concentrate to roughage = 70:30 in their study. We changed “7:3” to “70:30”, and changed “goat” to “Bore goat” to avoid confusion in the description of citation 55. Please see L446 for the details.
- Line 494: Do you mean HN or HC, or should you change the statement to avoid confusion?
au: Thanks for your kind suggestion. We agree with you and we have made changes to make the meaning of the sentence clearer, as detailed in L484~485.
- Tables 5 and 6 were highly advised to use the same format as Tables 3, 4, and 7.
au: Thanks for your suggestion. The format of Table 5 and 6 has been changed to match Tables 3, 4, and 7, and the page on which the tables are located has been changed to landscape so that the tables can be displayed completely on the page. Please see L294 Table 5 and L298 Table 6.
- Please correct Table 7. Something went wrong, so please correct it.
au: Thanks for your careful advice. We are very sorry that these problems have occurred in Table 7 due to our negligence. We have changed “Toal” to “Total” and revised the wrong top corner mark in the line of iso-valerate. Please see L311 Table 7 for details.
Again, many thanks for your valuable reviews.

Reviewer 2 Report
Dear Authors,
Thank you for revision. Very good work. I don't have no more suggestions and any objections.
Author Response
Dear Reviewer,
Thank you very much for your thoughtful suggestions that have helped improve this paper substantially. We really appreciate your approval of our revision. Moreover, we highly appreciate your time and consideration.
Sincerely,
Authors
- Thank you for revision. Very good work. I don't have no more suggestions and any objections.
au: Thanks for reviewer's recognition and positive comments on this study.
Again, many thanks for your valuable reviews.
